# Developing Production Guidelines for Baby Leaf Hemp (*Cannabis sativa* L.) as an Edible Salad Green: Cultivar, Sowing Density and Seed Size

**Renyuan Mi** [1] **, Alan G. Taylor** [2] **, Lawrence B. Smart** [2] **and Neil S. Mattson** [1,*]

[1] Horticulture Section, School of Integrative Plant Science, Cornell University, Ithaca, NY 14850, USA; rm974@cornell.edu

[2] Horticulture Section, School of Integrative Plant Science, Cornell AgriTech, Cornell University, Geneva, NY 14456, USA; agt1@cornell.edu (A.G.T.); lbs33@cornell.edu (L.B.S.)

[*] Correspondence: nsm47@cornell.edu; Tel.: +1-607-255-0621

**Abstract:** Scientific literature is lacking on cultural practices of baby leaf hemp production even though hemp (*Cannabis sativa* L.) is a widely grown crop for fiber and grain. The objective of this study was to develop a standard protocol to optimize yield and quality of baby leaf hemp production: cultivar screening, sowing density and seed size. Fresh weight (FW) and germination percentage was significantly affected by cultivars. Cultivars 'Picolo' and 'X-59' had a greater FW mainly due to greater germination percentage. In the sowing density experiment, 'Ferimon' and 'Katani' were evaluated at five seed densities, 0.65, 1.2, 1.75, 2.3 and 2.85 seeds·cm$^{-2}$ (42 to 182 seeds per cell). The FW and FW per plant (FWPP) had a positive quadratic response and negative quadratic response, respectively. Regarding seed size, cultivars 'Anka,' 'Ferimon' and 'Picolo' had the largest percentage of seeds, 26% to 30%, within the medium width size between 3.18 and 3.37 mm. Using the largest sized seeds (3.77 mm) increased FW by 34%, 26% and 23% as compared to non-sorted 'Anka', 'Ferimon' and 'Picolo' seeds, respectively. Overall, a greater understanding of cultivar selection, sowing density and seed-size distribution can promote greater yield and quality of baby leaf hemp as an edible salad green.

**Keywords:** baby leaf hemp; cultivar selection; sowing density; seed-size distribution

## 1. Introduction

*Cannabis sativa* L. has long been used for fiber, nutritional grain and medicinal purposes with an established history and widespread utilization. Hemp was recorded as a textile fiber used in China 6000 years ago [1]. Hemp was also found in Middle Eastern and Central Asia used for medicinal purposes [2] in antiquity. Since the start of the deregulation of hemp in the US by the 2014 Farm Bill, a rapid increase in the hemp production area arose in the US [3]. Hemp production tripled from 2017 (25,713 acres) to 2018 (78,176 acres) [4]. Since the 2018 Farm Bill, hemp was officially removed from schedule I of the Controlled Substance Act and hemp was defined as the *C. sativa* plant containing a tetrahydrocannabinol (THC) level less or equal than 0.3% [5]. Most state hemp pilot programs report that production acreage is primarily for cannabidiol (CBD) followed by grain and fiber.

Leafy greens are valued by consumers because they can offer high nutritional phytochemicals, fiber and mineral elements that humans need and also provide an appealing appearance, aroma, taste and texture to include in the daily diet [6]. Recently, baby leaf greens have increased in popularity with consumers over mature leaf greens because of their special flavor, freshness, convenience, and their bioactive compounds [7]. In the study by Xiao et al. [8], most young leaves of baby greens

had higher levels of phytochemicals than mature leaves. Currently, baby leaf hemp has been grown as a niche edible salad green. Anecdotally, we have observed a few producers in New York State that have been able to grow and sell their baby leaf hemp in New York City for up to $30 per pound. Based on the potential high-profit return, commercial growers are interested in baby leaf hemp production in controlled environment agriculture facilities (high tunnels, greenhouses and vertical farms). Growers usually harvest it at emergence of the third true leaf, which is approximately 12 to 18 d after sowing depending on the temperature and lighting condition. There is no published data on cultural requirements to support growers in achieving an efficient baby leaf hemp crop. After communicating with baby leaf hemp growers, it was found that their greatest concerns are the choice of cultivar and production methods.

Field hemp cultivars can be classified into four market classes: fiber, grain, medicinal and ornamentals [9], and breeding for improved cultivars has been conducted in all four market classes [10]. However, there is no scientific literature available on cultivar selection for baby leaf hemp. Information on the biomass yield and germination percentage of different cultivars is crucial for the success of commercial baby leaf hemp growers. Elite cultivars would not only offer maximum yield in a relative short period but would also provide uniformity and high quality which could bring higher profit for growers and lower the cost of production.

The effect of cultivar selection on yield is critical for successful production. For example, the length of the vegetative period of hemp was different between Ukrainian cultivars and French cultivars. In northern Europe, the advantage of early maturing cultivars is a shortened vegetative growth period, resulting in grain production before the onset of winter. However, for fiber production, northern Europe requires late maturing cultivars with longer vegetative growth period to maximize strong stem production [11]. By comparing nine cultivars from Ukraine, Hungary and France, the greatest single plot dry stem yield was collected from the Hungarian cultivars 'Kompolti,' 'Unico B' and French cultivar 'Futura 77' [12]. In the broader leafy greens literature, nine cultivars were evaluated by Grahn et al. [13] for determining the performance of extended season production in northwest Washington state. The greatest marketable yield was obtained by pak choi (*Brassica rapa* ssp. *chinensis* (L.) Hanelt.) 'Joi Choi' and mustard (*Brassica juncea* (L.) Czern.) 'Komatsuna'. For leaf lettuce (*Lactuca sativa* L. var. *crispa*), five cultivars ('Bergamo', 'Dubáček', 'Frisby', 'Lollo Rossa' and 'Redin') were investigated for yield. A significant effect of cultivar was exhibited in leaf head weight which varied from 164 to 502 g [14].

Field hemp is usually sown in high density to encourage rapid canopy closure and suppress weed growth [15]. High seeding rates also can promote yield and quality of hemp fiber by reducing branching and increasing the proportion of bast fiber content in the stem [16,17]. Bennett et al. [18] sowed field hemp in two densities, 0.015 and 0.03 seeds·cm$^{-2}$, and found greater yield and better weed control with the larger sowing density for all cultivars. However, no information is available on baby leaf hemp sowing density, leaving producers to question how to maximize yield. There is literature available on sowing density for common greenhouse vegetables such as microgreens or baby leaf greens. In a microgreen study, three microgreen cultivars were evaluated with five seed densities (1.1, 1.65, 2.2, 2.75 and 3.3 seeds·cm$^{-2}$), resulting in a quadratic increase in fresh weight (FW) and quadratic decrease in fresh weight per plant (FWPP) as sowing density increased from 1.1 to 3.3 seeds·cm$^{-2}$ [19]. The FW and FWPP were inversely correlated in a microgreen experiment evaluating three seeding rates (0.81, 1.62 and 2.37 seeds·cm$^{-2}$) [20]. Therefore, due to a wide variation in results based on species and harvest stage, the optimal sowing density for baby leaf hemp production should be investigated in order to provide an effective method for commercial growers.

Factors such as seed size are highly likely to affect the germination percentage and biomass yield of plants because small seeds contain less nutrition, which may lead to reduced seedling growth. Hemp seeds used in baby leaf production have typically been industrial hemp seed lots with non-sorted seed due to seed availability and cost. Germination and emergence of switchgrass increased nonlinearly as the seed size increased from five treatments (40, 50, 60, 70 and 80° air valve settings of a South Dakota seed blower) [21]. There is additional evidence that the higher germination percentage was

obtained with larger seeds compared to smaller seeds for *Erica vagans* L. A 90% germination percentage was attained from larger sized seeds with faster germination rate compared to smaller seeds [22]. A higher quality of seedlings was achieved with larger seed size of *Calluna vulgaris* L. [22]. The seed size of *Virola koschnyi* Aubl. 'Warb' did not significantly influence the germination percentage, but plant vigor was improved by larger seeds [23]. In general, a larger seed mass was demonstrated to produce more vigorous plants due to a more developed embryo and larger energy reserves [24].

In our work, we consider baby leaf hemp as a potential crop in the greenhouse environment grown on a soilless substrate and with liquid fertilizer. In this regard, baby leaf hemp could be grown year-round in a consistent manner. The objective of our work is to determine the impact of cultivar, seeding density and seed size distribution on the yield and quality (morphology, emergence) on hemp seedlings with a goal of optimizing procedures for commercial production in controlled environments.

## 2. Materials and Methods

Seed of nine hemp cultivars was obtained from those entered in Cornell University (Ithaca, NY, USA) 2018 field trials and grown under pilot program research authorization from NYS Dept of Agriculture and Markets. Of these nine cultivars, five were dual-purpose (D) and four were for dedicated grain cultivars (G). Nine cultivars were purchased from UNISeeds (Cobden, ON, Canada), Assocanapa USA (Lexington, KY, USA), HGI (Saskatoon, SK, Canada), Legacy Hemp (Hastings, MN, USA) and Parkland (Dauphin, MB, Canada) (Table 1). Prior to our experiments, there was no available information on flavor, aroma or other quality attributes of baby leaf hemp. Therefore, in consultation with commercial growers, the initial selection of these nine cultivars was based on cost (including seed price and shipping) as well as availability. At the time of our experiment, the lowest cost cultivar was 'Anka' ($5.83/kg) and the most expensive cultivars were 'Canda' and 'Joey' (each at $22.11/kg). For all experiments the following common methods were used. Plants were grown in a single layer glass greenhouse located at Cornell University in Ithaca, NY (42° N latitude) under ambient light and placed on a bench made by galvanized steel elevated 85 cm from the floor. Each experimental unit was planted in a polystyrene cell ($8 \times 8 \times 6$ cm; Dillen-ITML Greenhouse, Twinsburg, OH, USA) placed on the bench. For all experiments, the substrate mix was a custom seeding mix (Jiffy Group, Zwijndrecht, Zuid-Holland, The Netherlands), which was a blend of OMRI (Organic Materials Review Institute) approved coconut coir, peat moss from Jiffy Canada (Lorain, OH, USA) and dolomitic limestone. Substrate nutrient analysis was conducted (J.R. Peter's Inc., Allentown, PA, USA) with the following values: 0.34 ppm nitrate ($NO_3$-N), 2 ppm of ammonium ($NH_4$-N), 2.31 ppm phosphorus (P), 82 potassium (K), 6 ppm calcium (Ca), 10 ppm magnesium (Mg), 4 ppm sulfur (S), 0.04 ppm boron (B), 0.24 ppm iron (Fe), 0.01 ppm manganese (Mn), 0.03 ppm copper (Cu), 0.01 ppm zinc (Zn), 0.08 ppm molybdenum (Mo), 0.12 ppm aluminum (Al), 35 ppm sodium (Na) and 179 ppm chloride (Cl). The initial substrate pH was 5.58 and EC (Electrical Conductivity) was 0.63 dS·m$^{-1}$. The substrate mix was prepared by mixing it with RO (reverse osmosis) water in a 2:1 ratio by volume to achieve adequate moisture. Cells were filled to a 5 cm substrate depth, seeds were sown and an additional 1 cm of the same substrate was covered on the top of the seeds. All treatments received the same access to irrigation water with water soluble fertilizer by sub-irrigation by filling a flat to a 3 cm level with a 150 mg·L$^{-1}$ N nutrient solution (21 N-2.2 P-16.6 K Jack's All-Purpose Liquid Feed, J.R. Peter's Inc., Allentown, PA, USA) and allowing each cell to take up water for 90 s before removing. The germination period (time to seedling emergence) usually took 48 to 96 h depending on the temperature and lighting conditions. Plants were harvested when half of the seedlings reached the stage of emergence of the third true leaf. The time period from seed to harvest was around 13 to 18 d depending on the temperature and lighting conditions. Temperature and relative humidity during each experiment and crop cycle are listed in Table 2. In the experiments, an experimental unit was considered to be one $8 \times 8$ cm cell. Measurements were collected for germination percentage (number of seedlings emerged divided by total sown seeds), height (from the surface of the substrate to the tallest part of representative seedlings), and fresh weight (FW, using only the epicotyl, i.e., part of the plant above

the cotyledons, based on commercial practice). After harvesting for FW, epicotyls from all plants in an experimental unit were bagged and placed in a 70 °C oven for 72 h to determine dry weight (DW). The fresh weight per plant (FWPP) was calculated as FW divided by number of seedling emergence for each treatment.

**Table 1.** Nine hemp cultivars used in this study with their sources and seed lot number.

| Cultivars | Source | Seed Lot Number |
|-----------|--------|-----------------|
| 'Anka' | UNISeeds | 4371-6032 |
| 'Canda' | Parkland | 828-17-04 |
| 'Ferimon' | UNISeeds | F1545X64110 |
| 'Joey' | Parkland | 828-17-18S |
| 'Katani' | HGI | 15-DPKARE-01 |
| 'Picolo' | HGI | 980-8-16KEFR-PICE-01 |
| 'USO-31' | Assocanapa USA | F1545R154001B |
| 'Wojko' | Assocanapa USA | MH/2914/WOJ1 |
| 'X-59' | Legacy Hemp | 2208.8 |

**Table 2.** Average Daily Temperature (ADT, °C) and Average Daily Relative Humidity (ADRH, %) in three cycles of each experiment.

| Experiment | Crop Cycle | ADT (°C) | ADRH (%) |
|------------|-----------|----------|----------|
| Cultivar selection | 1 | 20 | 60 |
| | 2 | 19 | 66 |
| | 3 | 22 | 76 |
| Sowing density | 1 | 22 | 76 |
| | 2 | 23 | 80 |
| | 3 | 23 | 81 |
| Seed size | 1 | 19 | 41 |
| | 2 | 19 | 46 |
| | 3 | 19 | 37 |

## 2.1. Cultivar Selection Experiment

Cultivars 'Anka', 'Katani', 'Ferimon', 'Wojko', 'USO-31', 'X-59', 'Picolo', 'Canda' and 'Joey' were evaluated in this experiment (Figure 1). Seeds of these nine different cultivars were sown evenly on the substrate and covered with 1 cm of the same substrate with a sowing density at 1.2 seeds·cm$^{-2}$ (i.e., 77 seeds per cell). For each crop cycle there were six experimental units. The time of the three crop cycles from seed sowing until harvest were 16th April to 29th April, 30th April to 13rd May and 18th June to 1st July, 2019, respectively. Plants were harvested when the third true leaf emerged for at least 50% of the plants, and we measured FW, DW, height, and germination percentage (as described above).

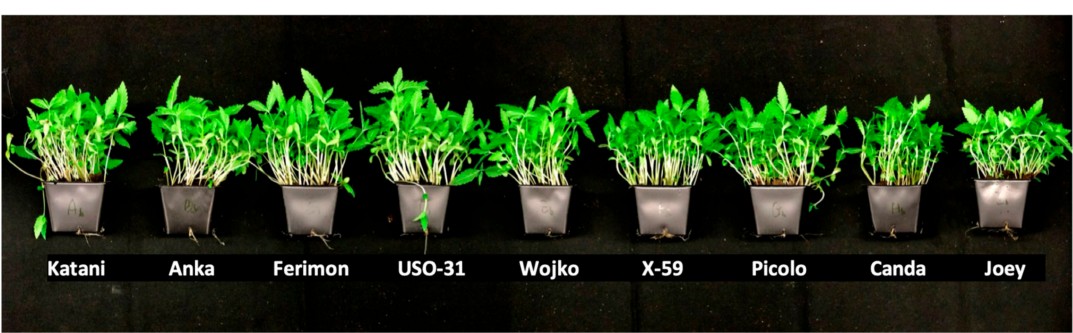

**Figure 1.** Typical appearance of nine baby leaf hemp at harvest stage (front view).

### 2.2. Sowing Density Experiment

Cultivars 'Ferimon' and 'Katani' were selected to evaluate sowing density of baby leaf hemp at five different seeding rates: 0.65, 1.2, 1.75, 2.3 and 2.85 seeds·cm$^{-2}$ (i.e., 42, 77, 112, 147 and 182 seeds·cell$^{-1}$). The experiment was repeated over three crop cycles which took place: 19th June to 2nd July, 9th July to 20th July and 20th July to 31st July, 2019. Plants were harvested at the third true leaf stage and all parameters described above with FWPP were recorded for each experimental unit.

### 2.3. Seed Size Experiment

Cultivars 'Anka', 'Picolo' and 'Ferimon' were selected to evaluate the seed-size distribution and the effect of seed size on measured parameters. Seeds were sorted by size using eight different 23 cm × 23 cm hand screen sieves (Seedburo Equipment Company, Des Plaines, IL, USA) with round perforations for sorting by seed width. Seeds were sized through a series of stacked sieves from largest to smallest and seeds retained on a particular sieve were grouped: 3.77 mm (i.e., sieve size 9.5/64 inches), 3.57 mm (9/64 inches), 3.37 mm (8.5/64 inches), 3.18 mm (8/64 inches), 2.98 mm (7.5/64 inches), 2.78 mm (7/64 inches), 2.58 mm (6.5/64 inches) and 2.83 mm (6/64 inches). The distribution of 100 g of seed from each cultivar according to the sieve sizes was collected and analyzed (in terms of percent of seeds in each class by weight) with three replicates. In the evaluation of the effect of seed size on biomass/yield of baby leaf hemp, there were seven treatments according to seed width size: control (non-sorted seeds), sieve size 3.77, 3.57, 3.18, 2.98 and <2.98 mm. Sieved seeds of three entries from each treatment were sown evenly at a density 0.47 seeds·cm$^{-2}$. The experiment was repeated three times with seeding date and harvesting dates of: 8 December to 26 December 2019; 30 December 2019, to 17 January 2020; and 21 January to 8 February 2020, respectively. Plants were harvested at the third true leaf stage and all parameters mentioned above were recorded for each experimental unit.

### 2.4. Statistical Analysis

In cultivar selection experiment, for each crop cycle there were six blocks placed in a randomized complete block design where each block contained one experimental unit from each of the nine different cultivars. In the sowing density experiment, there were five experimental units per cultivar and sowing density treatments arranged in a randomized complete block design where each block consisted of one experimental unit from each cultivar at each of the five seed densities. Within a block, the 10 cells were completely randomized. For seed size experiment, each crop cycle had five blocks where each block consisted of one experimental unit per cultivar per seed size treatment. The experiment was also arranged in a randomized complete block design. The block was based on location in the greenhouse bench. All three experiments were replicated over time for a total of three crop cycles. Data were analyzed with R studio (Version 1.2.1335, RStudio, Inc., Boston, MA, USA), using a mixed model including linear and quadratic regression (when treatments followed a quantitative independent variable, i.e., sowing density), Analysis of Variance (ANOVA), and mean separation comparison by Tukey's honestly significant difference test (alpha = 0.05) with the following packages in R studio: library(ggplot2), library(multcomp), library(emmeans), library(lsmeans), library(lme4).

## 3. Results

### 3.1. Cultivar Selection Experiment

There was a significant difference for all measured parameters in response to cultivar, crop cycle, and cultivar × crop cycle interaction but there was no significant difference due to block except for fresh weight (Table 3). There was no significant cultivar × block interaction, except dry weight (Table 3).

**Table 3.** Analysis of Variance on measured parameters for baby leaf hemp.

| ANOVA Table | | | | | |
|---|---|---|---|---|---|
| Main Effects & Interactions | Germination | Height | Fresh Weight | Dry Weight | Fresh Weight Per Plant |
| Cultivar | *** | *** | *** | *** | *** |
| Block | NS | NS | * | NS | NS |
| Crop Cycle | *** | *** | *** | *** | *** |
| Cultivar × Block | NS | NS | NS | * | NS |
| Cultivar × Cycle | *** | ** | ** | ** | * |

NS, *, **, *** Nonsignificant or significant at $p \leq 0.05$, 0.01 or 0.001, respectively.

There was a significant difference observed between the nine cultivars for germination percentage (Table 3), which ranged from 51% ('Wojko') to 81% ('Picolo') (Table 4). 'Picolo', 'X-59' and 'Ferimon' had a significantly greater germination percentage than 'USO-31', 'Canda', 'Wojko' and 'Joey'. There were no significant differences between 'Anka' and 'Katani' for germination percentage, but they had a significantly lower germination percentage than 'Picolo' and 'X-59'. Therefore, regarding germination, 'Picolo' and 'X-59' had good germination capacity among these nine cultivars for the seed lots tested in this experiment.

**Table 4.** Growth parameters of nine baby leaf hemp cultivars. Data represent means of 18 experimental units (3 crop cycles each with 6 experimental units per treatment).

| Cultivars | Germination Percentage (%) | Height (cm) | Fresh Weight (g·cell$^{-1}$) | Dry Weight (g·cell$^{-1}$) | Fresh Weight Per Plant (g·plant$^{-1}$) |
|---|---|---|---|---|---|
| 'Anka' | 67 [c,d] | 12.4 [b] | 8.6 [b,c] | 1.0 [c,d] | 0.17 [b] |
| 'Canda' | 58 [e,f] | 11.8 [b] | 8.7 [b,c] | 1.1 [b,c] | 0.20 [a] |
| 'Ferimon' | 72 [b,c] | 12.3 [b] | 9.7 [a,b] | 1.1 [a–c] | 0.18 [a,b] |
| 'Joey' | 53 [f] | 11.5 [b] | 6.9 [d] | 0.8 [e] | 0.17 [a,b] |
| 'Katani' | 70 [b–d] | 13.7 [a] | 9.7 [a,b] | 1.2 [a,b] | 0.18 [a,b] |
| 'Picolo' | 81 [a] | 12.5 [b] | 10.3 [a] | 1.3 [a] | 0.16 [b] |
| 'USO-31' | 64 [d,e] | 12.5 [b] | 8.9 [b] | 1.0 [b,c] | 0.19 [a,b] |
| 'Wojko' | 51 [f] | 11.8 [b] | 7.5 [c,d] | 0.8 [d,e] | 0.19 [a] |
| 'X-59' | 75 [a,b] | 12.0 [b] | 9.8 [a,b] | 1.8 [a,b] | 0.17 [b] |

Letters represent mean separation comparison using Tukey's HSD (alpha = 0.05).

There was no significant difference between the nine cultivars for height except 'Katani', which was significantly taller than other cultivars. The range of height of these nine cultivars was between 11.5 and 13.7 cm (Table 4). A significant difference between the nine cultivars was observed for FW, which ranged from 6.9 g·cell$^{-1}$ to 10.3 g·cell$^{-1}$ (Table 4). 'Picolo' had the greatest FW and was 33% larger than 'Joey', which had the smallest FW. 'Picolo' had a significantly greater FW than five cultivars ('USO-31', 'Canda', 'Ank', 'Wojko' and 'Joey'). 'X-59,' 'Katani', 'Ferimon' and 'USO-31' had a significantly larger FW than 'Wojko' or 'Joey'. The FW (per cell) was closely related to germination percentage, so germination percentage seems to be an important attribute for obtaining a good yield.

The DW results closely followed FW. Dry weight varied from 0.78 to 1.27 g·cell$^{-1}$ between the nine cultivars (Table 4). The greatest DW was for 'Picolo', which was 39% greater than the lowest cultivar, 'Joey.' 'Picolo', 'X-59' and 'Katani' exhibited the greatest DW and these were significantly greater than 'Anka', 'Wojko' and 'Joey'.

The range of FWPP among nine cultivars was from 0.163 to 0.196 g·plant$^{-1}$ (Table 4). 'Canda' and 'Wojko' had a significantly larger FWPP than 'X-59', 'Anka' and 'Picolo'. No significant difference in FWPP was observed between 'USO-31', 'Katani', 'Ferimon' and 'Joey'.

### 3.2. Sowing Density Experiment

In the investigation of optimal sowing density on baby leaf hemp production, no significant effect of sowing density was observed on the germination of 'Ferimon' or 'Katani'. In general, 'Ferimon' had a significantly greater germination percentage than 'Katani'. The mean germination percentage of 'Ferimon' and 'Katani' was 69% and 63%, respectively.

There was no significant effect of sowing density on height of 'Katani' or 'Ferimon'. Overall, 'Katani' was slightly taller (mean of 14.7 cm across treatments) than 'Ferimon' (mean of 13.3 cm across treatments) (Table 5). FW of both 'Katani' and 'Ferimon' showed a positive response to sowing density, whereby FW increased quadratically as sowing density increased from 0.65 to 2.85 seeds·cm$^{-2}$ (Figure 2; Table 6). There was no significant interaction between cultivar and sowing density (Table 5). The increasing FW response to sowing density began to plateau at the two greatest densities (Figure 2). For example, FW of 'Katani' increased 28% as density increased from 0.65 to 1.2 seeds·cm$^{-2}$, but increased by only 11% as density increased from 2.3 to 2.85 seeds·cm$^{-2}$. Similarly, for 'Ferimon', FW increased 27% as density increased from 0.65 to 1.2 seeds·cm$^{-2}$, but increased by only 7% as density increased from 2.3 to 2.85 seeds·cm$^{-2}$.

**Table 5.** Analysis of Variance for sowing density parameters and model-accounted variability $R^2$.

| ANOVA Table | | | | | |
|---|---|---|---|---|---|
| Main Effect and Interaction | Germination | Plant Height | Fresh Weight | Dry Weight | Fresh Weight Per Plant |
| Density | NS | NS | *** | *** | *** |
| Density$^2$ | NS | NS | ** | NS | ** |
| Cultivar | * | * | NS | NS | ** |
| Cultivar × Density | NS | NS | NS | NS | NS |
| $R^2$ | 0.193 | 0.843 | 0.926 | 0.888 | 0.869 |

NS, *, **, *** Nonsignificant or significant at $p \leq 0.05$, 0.01 or 0.001, respectively. $R^2$ represented model-accounted variability for each parameter.

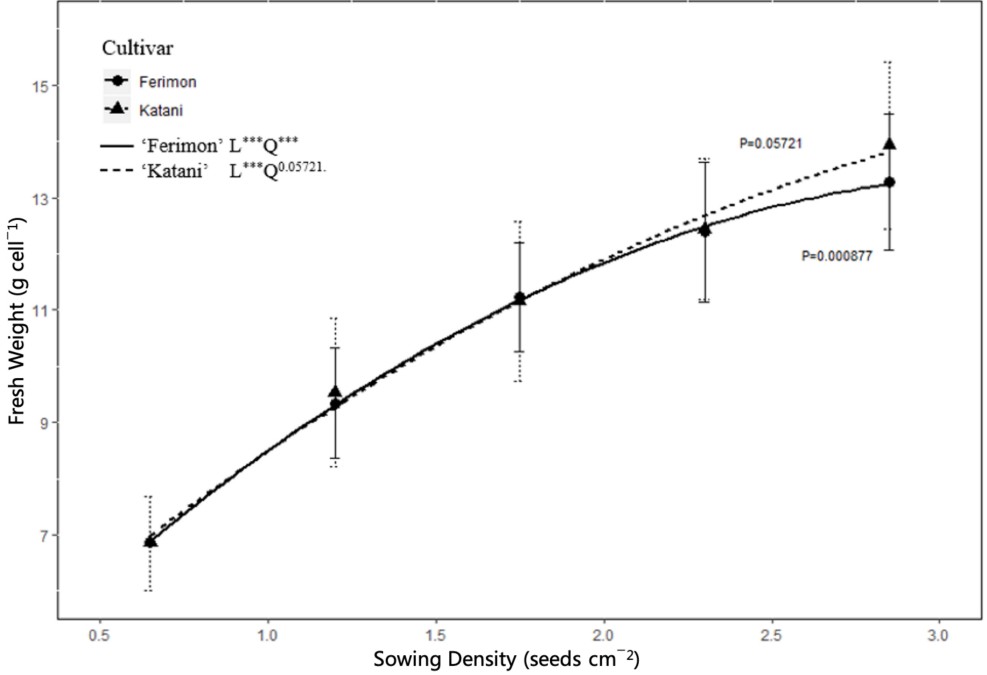

**Figure 2.** Fresh weight per cell of baby leaf hemp 'Ferimon' and 'Katani' in response to increasing sowing density. Data represent mean ± SE (Standard Error) of 15 experimental units (3 crop cycles each with 5 experimental units per treatment per cultivar). Significance of linear (L) and quadratic (Q) regression represented as ***, significant at $p \leq 0.001$.

**Table 6.** Regression values for growth parameters by sowing density for 'Katani' and 'Ferimon', including squared terms for a quadratic model.

| Coefficients | Estimate [A] | Std. Error [A] | *p*-Value [B] |
|---|---|---|---|
| **Fresh Weight** | | | |
| *'Katani'* | | | |
| Intercept | 11.6 | 1.10 | *** |
| Density | 5.3 | 1.18 | *** |
| Density$^2$ | −0.641 | 0.330 | 0.057 |
| *'Ferimon'* | | | |
| Intercept | 11.0 | 0.876 | *** |
| Density | 6.11 | 0.933 | *** |
| Density$^2$ | −0.919 | 0.262 | *** |
| **Dry Weight** | | | |
| *'Katani'* | | | |
| Intercept | 1.26 | 0.149 | *** |
| Density | 0.641 | 0.158 | *** |
| Density$^2$ | −0.068 | 0.044 | NS |
| *'Ferimon'* | | | |
| Intercept | 1.21 | 0.133 | *** |
| Density | 0.669 | 0.142 | *** |
| Density$^2$ | −0.085 | 0.040 | * |
| **Fresh Weight Per Plant** | | | |
| *'Katani'* | | | |
| Intercept | 0.501 | 0.028 | *** |
| Density | −0.195 | 0.030 | *** |
| Density$^2$ | 0.035 | 0.008 | *** |
| *'Ferimon'* | | | |
| Intercept | 0.436 | 0.022 | *** |
| Density | −0.129 | 0.023 | *** |
| Density$^2$ | 0.020 | 0.007 | *** |

NS, *, **, *** Nonsignificant or significant at $p \leq 0.05$, 0.01 or 0.001, respectively. [A] Estimated response when all other treatment effects are equal to zero [B] Significance when treatment effects are at their average value.

For DW, a similar pattern was found as in FW, in which there was also a positive response to sowing density. 'Ferimon' displayed both significant linear and quadratic response to DW as the sowing density increased (Figure 3; Table 6). The increase of DW as sowing density increased also plateaued at the maximum sowing density. For example, DW of 'Ferimon' increased 37% as sowing density increased from 0.65 to 1.2 seeds·cm$^{-2}$, but DW only had a 9% increase as sowing density increased from 2.3 to 2.85 seeds·cm$^{-2}$. However, for 'Katani', the linear regression represented a better fit than a quadratic regression (Figure 3). For 'Katani', DW increased from 0.85 to 1.79 g as sowing density increased from 0.65 to 2.85 seeds·cm$^{-2}$.

Both cultivars had a significant response of FWPP to sowing density. No significant interaction between cultivar and sowing density occurred, but there was a significant difference between cultivars (Table 5). 'Katani' had a significantly larger FWPP than 'Ferimon'. There was a significant quadratic decrease in FWPP of both 'Katani' and 'Ferimon' as the sowing density increased from 0.65 to 2.85 seeds·cm$^{-2}$ (Figure 4; Table 6). For example, for 'Katani', the FWPP decreased 29%, 22%, 17% and 11% when the sowing density increased from 0.65 to 1.2, 1.2 to 1.75, 1.75 to 2.3 and 2.3 to 2.85 seeds·cm$^{-2}$, respectively. For 'Ferimon', the FWPP decreased 25%, 17%, 16% and 15% as the sowing density increased from 0.65 to 1.2, 1.2 to 1.75, 1.75 to 2.3 and 2.3 to 2.85 seeds·cm$^{-2}$, respectively.

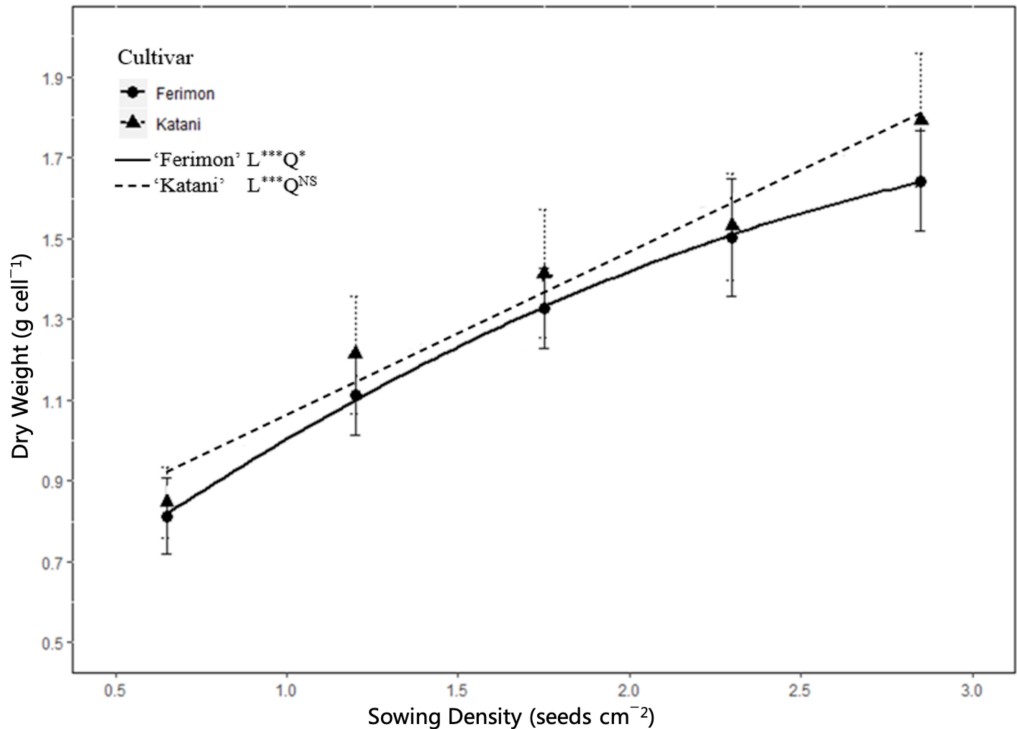

**Figure 3.** Dry weight per cell of baby leaf hemp 'Ferimon' and 'Katani' in response to increasing sowing density. Data represent mean ± SE of 15 experimental units (3 crop cycles each with 5 experimental units per treatment per cultivar). Significance of linear (L) and quadratic (Q) regression represented as NS, *, ***, Nonsignificant or significant at $p \leq 0.05$ or 0.001, respectively.

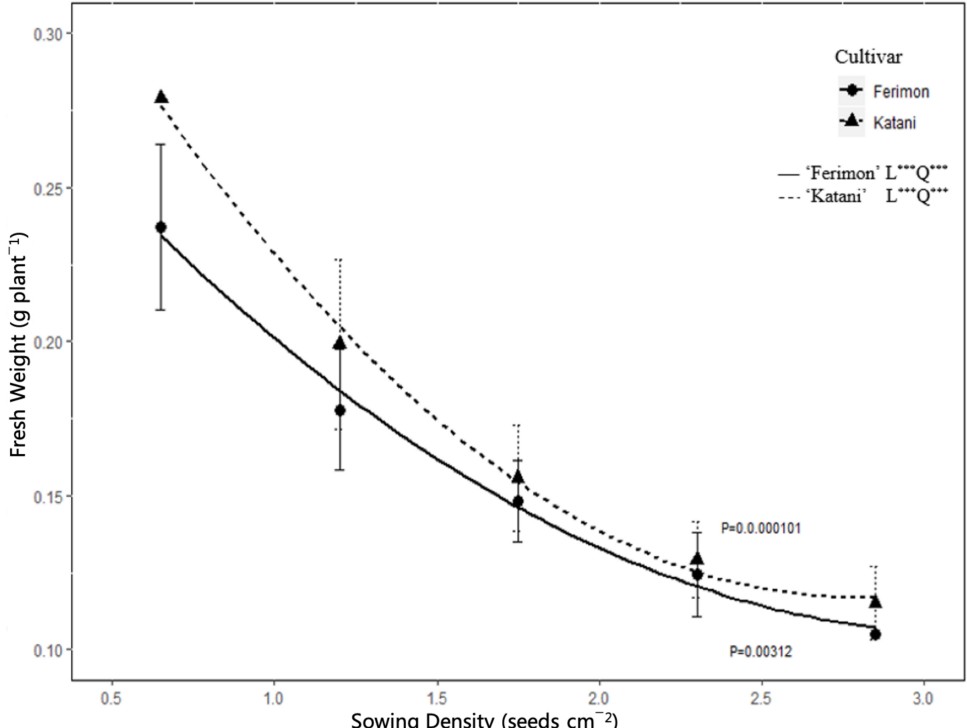

**Figure 4.** Fresh weight per plant of baby leaf hemp 'Ferimon' and 'Katani' in response to increasing sowing density. Data represent mean ± SE of 15 experimental units (3 crop cycles each with 5 experimental units per treatment per cultivar). Significance of linear (L) and quadratic (Q) regression represented as ***, significant at $p \leq 0.001$.

### 3.3. Seed Size Experiment

To determine the effect of seed size and distribution on yield and quality of baby leaf hemp seedlings, we evaluated the seed-size distribution. It exhibited a similar pattern for all three cultivars studied. The greatest percentage of seeds (by weight) were in the category of sieve size 3.18 and the lowest percentage of seeds were in the category of sieve size <2.38 mm width (Figure 5). Seeds with sieve size greater than 2.78 mm width represented 90% to 94% of all seeds.

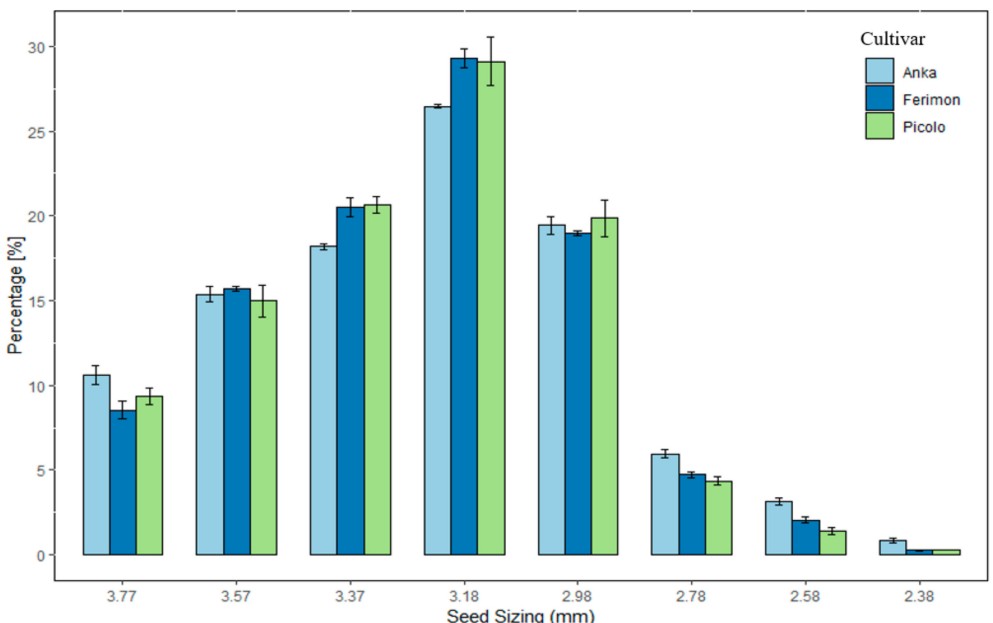

**Figure 5.** Hemp seed-size distribution for 'Anka', 'Ferimon' and 'Picolo'. Data are means ± SE of three seed lots (each consisting of 100 g) per cultivar.

A significant effect of seed size treatment and cultivar was observed for all growth parameters (Table 7). There were significant interactions between seed size and cultivar for FW and Height. There were also significant effects of block and cycle for all parameters except germination.

**Table 7.** Analysis of Variance for seed size parameters.

| ANOVA Table | | | | | |
|---|---|---|---|---|---|
| Main Effects & Interactions | Germination | Height | Fresh Weight | Dry Weight | Fresh Weight Per Plant |
| Treatment | *** | *** | *** | *** | *** |
| Cultivars | *** | *** | *** | *** | *** |
| Block | NS | * | *** | *** | *** |
| Cycle | NS | *** | *** | *** | *** |
| Treatment × Cultivars | NS | ** | * | NS | NS |
| Treatment × Block | NS | NS | NS | NS | NS |
| Treatment × Cycle | NS | NS | NS | NS | * |

NS, *, **, *** Nonsignificant or significant at $p \leq 0.05$, 0.01 or 0.001, respectively.

The effect of seed size was shown for each parameter of each cultivar [25]. Averaged for the three cultivars, seed size <2.98 mm had the lowest germination compared to all other sizes and not the non-sorted control (Figure 6). The largest sized seed fractions had greater dry weight and fresh weight per plant, while the smallest size fractions had lower dry weight and fresh weight per plant compared to the control (Figure 6). For height, a significant interaction between seed size treatments and cultivars was found, and the height of 'Anka', 'Ferimon' and 'Picolo' ranged from 7.07 to 10.0,

8.6 to 10.3 and 8.5 to 10.4 cm, respectively (Table 7; Figure 7). For 'Anka' and 'Ferimon', the largest seed size of 3.77 mm was significantly taller than non-sorted treatment, but not for 'Picolo'. Only seed size less than 2.98 mm had a shorter height than non-sorted treatment among three cultivars (Figure 7).

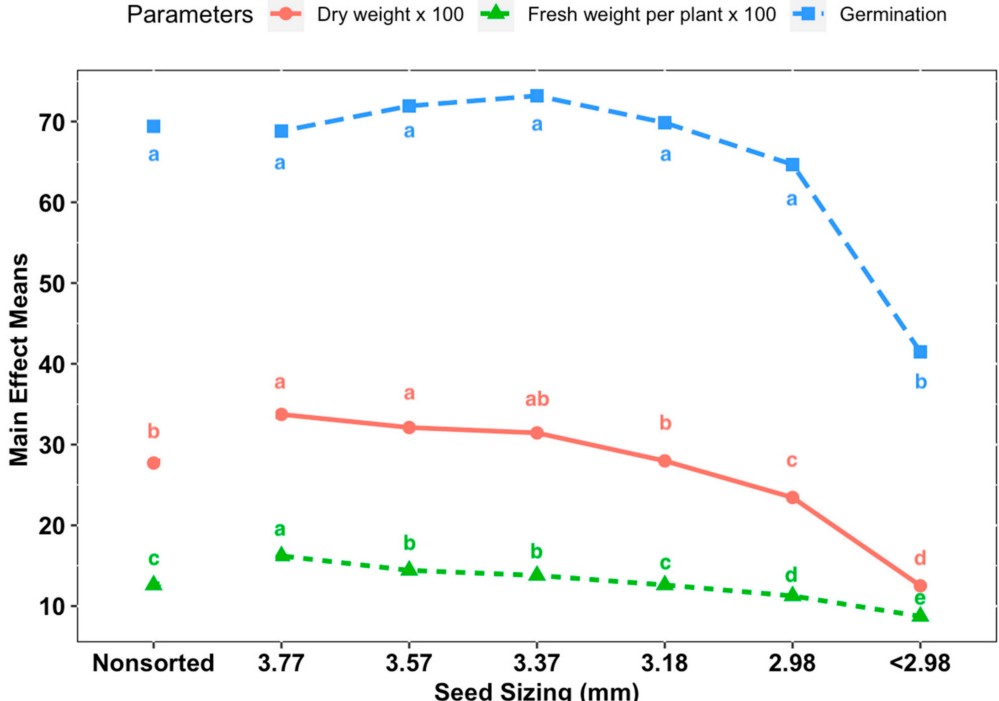

**Figure 6.** Main effect means of growth parameters for 'Anka', 'Ferimon' and 'Picolo' in response to seed size treatment. Data represent mean of 15 experimental units (3 crop cycles each with 5 experimental units per treatment). Letters represent mean separation comparison using Tukey's HSD (alpha = 0.05).

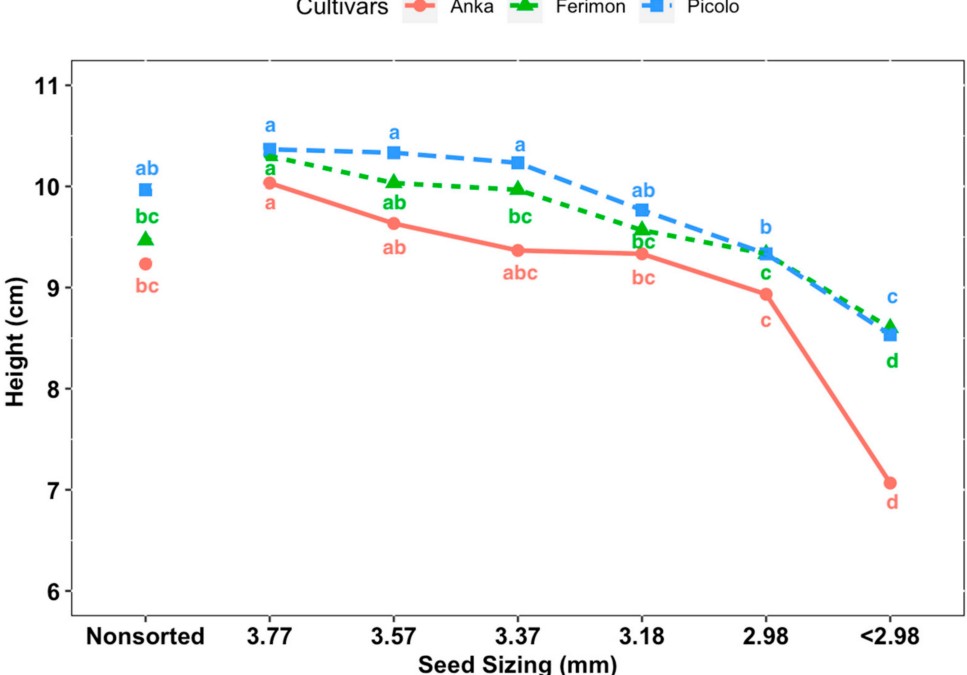

**Figure 7.** Height of baby leaf hemp cultivars 'Anka', 'Ferimon' and 'Picolo' in response to seed size treatment. Data represent mean of 15 experimental units (3 crop cycles each with 5 experimental units per treatment). Letters represent mean separation comparison using Tukey's HSD (alpha = 0.05).

Overall, there was a pattern of declining FW as seed size decreased. 'Ferimon' had the greatest FW among three cultivars. 'Ferimon' and 'Picolo' showed a consistent trend in FW with respect to seed size effect (Figure 8). However, 'Anka' had an unexpectedly lower FW for seed size 3.57 mm. Seed size 3.77 and 3.57 mm had a significantly larger FW than non-sorted seeds for 'Ferimon' and 'Picolo', but for 'Anka', only seed size 3.77 mm had a significantly larger FW than the non-sorted treatment. The largest-sized seeds (3.77 mm) had a FW that was 35%, 27% and 23% greater than non-sorted seeds for 'Anka', 'Ferimon' and 'Picolo,' respectively (Figure 8). For 'Ferimon' and 'Picolo', seed size less than 2.98 mm had a significantly smaller FW than non-sorted treatment, but for 'Anka', seed size of <2.98 or 2.98 mm had a significantly smaller FW than non-sorted treatment. 'Anka,' 'Ferimon' and 'Picolo' had FW reductions of 71%, 55% and 49% for the smallest sized seeds, respectively, relative to non-sorted seeds.

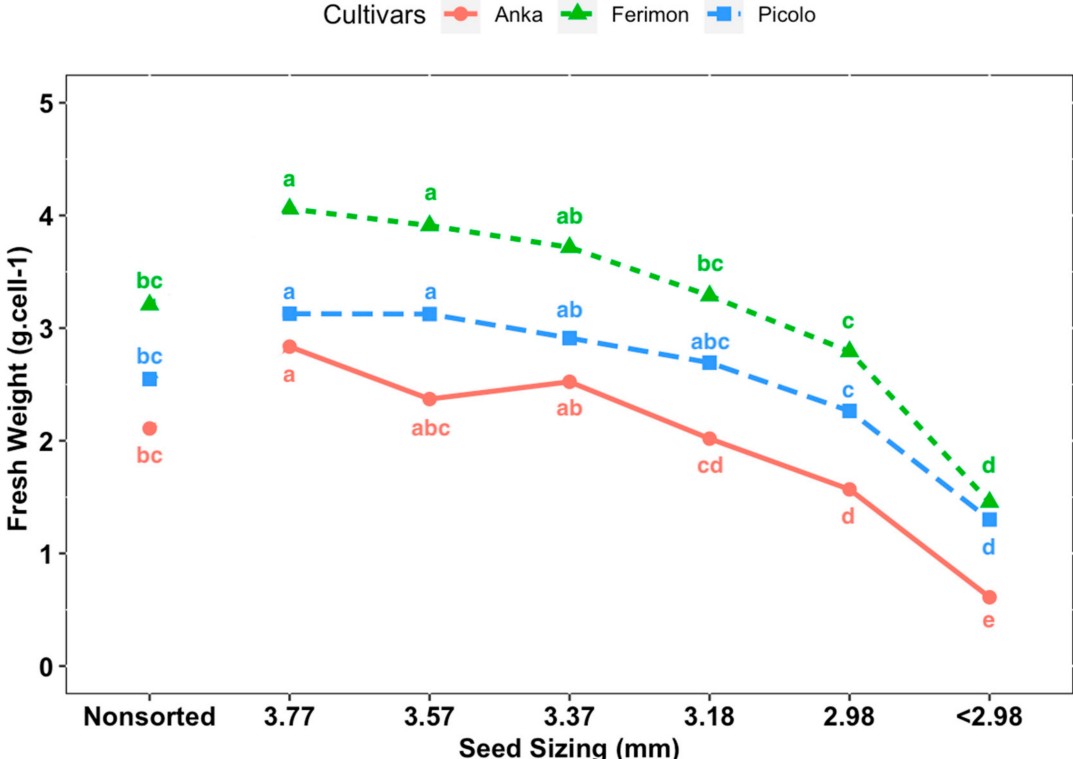

**Figure 8.** Fresh weight of baby leaf hemp cultivars 'Anka', 'Ferimon' and 'Picolo' in response to seed size treatment. Data represent mean of 15 experimental units (3 crop cycles each with 5 experimental units per treatment). Letters represent mean separation comparison using Tukey's HSD (alpha = 0.05).

## 4. Discussion

### 4.1. Cultivar Selection Experiment

The objective of cultivar selection was to determine which field hemp dual or grain cultivars produce the greatest quality (germination and FW) when grown as an edible salad green. 'Anka' was used as an industry standard cultivar for a number of field hemp experiments by our colleagues at Cornell University due to seed availability and cost, but the seed lot we used showed poor performance in germination. In our study, we found that germination percentage was highly correlated with FW ($p = 0.0002$, $R^2 = 0.858$) and DW ($p = 0.0005$, $R^2 = 0.816$) among the nine cultivars for the seed lots tested. Because the baby leaf hemp production cycle is short (to reach emergence of the third true leaf), the yield performance was highly related to the germination percentage of each cultivar. We propose that each sown seed producing a vigorous seedling contributes more to the final yield than other factors such as height of the plant. Therefore, for commercial baby leaf hemp growers, selection of cultivars

with good germination is critical for success. In our study, 'Picolo', 'X-59', 'Ferimon' and 'Katani' were good performing cultivars based on germination and subsequent yield. Regarding cultivar selection, while there is no previous research with baby leaf hemp, other studies with baby leaf greens or microgreens illustrated that cultivar selection has a large effect on yield enhancement [13,14]. In an examination of 10 microgreen species, there was a large variability in germination time with 1 to >14 d required to reach 75% germination as well as a large range of germination percentage between 10% to 98% under 12 h light and dark, respectively. In another example, the yield of perennial wall rockets (*Diplotaxis tenuifolia* (L.) DC.) was affected by cultivars, which illustrates the importance of cultivar studies for optimizing marketable weight [11]. In fiber hemp, Lisson and Mendham [12] determined cultivars for the greatest single plot dry stem yield. Except for cultivar selection for yield, the success of growing fiber hemp also depended on the selection of cultivars less sensitive to photoperiod and cultivation on well drained sites [12]. In our study, selecting elite cultivars with high quality of germination and fresh weight yield can contribute to optimization of baby leaf hemp production practices. While we determined response in terms of yield and germination, other attributes such as sensory preferences and nutritional values were impacted by cultivar [25]. Therefore, both best production practices and consumer preferences need to be considered to achieve success in the baby greens market. There could be a variation of quality between seed lots in one cultivar. In my experiment, the limitation of time only allowed me to repeat the experiment three times with the same seed lot for each cultivar. In future work, more evaluation of these cultivars with different seed lots should be conducted to further confirm the results.

### 4.2. Sowing Density Experiment

Usually commercial seed companies provide recommended seeding rates for growers, but because baby leaf hemp is a new niche crop, there is currently no information on this subject. Although the research by Bennett et al. [18] used seeding rates for fiber hemp in low and high seed rates, 0.015 and 0.03 seeds·cm$^{-2}$, respectively, our seeding rates were 22 (0.65 seeds·cm$^{-2}$) to 95 (2.85 seeds·cm$^{-2}$) times greater. Therefore, we adapted information from other baby leaf green species such as arugula, mizuna (*Brassica rapa nipposinica*—(L.H. Bailey.) Hanelt.) and mustard, which were sown at a seeding rate from 1.1 to 3.3 seeds·cm$^{-2}$. The FW of these microgreens also demonstrated a quadratic increase in yield as sowing density increased [19] with a similar pattern whereby diminishing increases in FW/DW were observed at the highest densities. In the evaluation of the microgreen table beet (*Beta vulgaris* L.), the commercially recommended sowing density 201 g·m$^{-2}$ could lead to a greater shoot fresh weight per m$^2$ than treatments with lower seeding rates [20]. However, a negative aspect found by Murphy et al. [20] was that at higher sowing density resulted into a lower biomass shoots, which is similar to our finding of lower FWPP as density increases. In our experiment, we found that density did not affect height, but since it affected FWPP, at higher density we have thinner and lighter plants that are more prone to lodging. Three microgreen species in a culinary assessment study were utilized at a sowing density of 3 seeds·cm$^{-2}$ [26]. Because the hemp seeds were slightly larger than these Brassicaceae microgreen seeds, we used sowing density from 0.65 to 2.85 seeds·cm$^{-2}$. While 2.85 seeds·cm$^{-2}$ led to the greatest FW/DW in our study (Figures 2 and 3), we did observe an elevated susceptibility of disease (gray-colored mold on the surface of substrate) in the maximum seeding rate which was not reflected in the data. Hemp is very susceptible to diseases such as gray mold (*Botrytis cinerea*), hemp canker (*Fusarium* spp.) and damping off (mostly caused by *Pythium* spp.) [27]. Therefore, based on yield results of our experiment, the recommended sowing density for commercial growers would be 2.3 to 2.85 seeds·cm$^{-2}$. We observed that increased sowing density extended the time period to harvest. We harvested all plants at the same time during the third true leaf emergence for the average of plants in all treatments, but it was obvious that with a lower sowing density, plants tended to grow faster and thicker with more true leaves formed. Future work should examine the effect of sowing density on seeding development rate, further measures of quality (stem

thickness, disease incidence) and the interaction with other cultural factors (for example, at higher density greater air circulation may be required to prevent foliar-borne pathogens).

*4.3. Seed Size Experiment*

For conventional baby leaf and microgreens crops, seed companies typically sell seed lots of uniform size for commercial growers. However, little seed sizing was observed in the hemp seed lots we used, possibly as this is a new product niche. Based on our observation of previous experiments, the performance of baby leaf hemp including biomass yield and germination was impacted by different sized seeds. Smaller seeds also tend to produce more abnormal seedlings or result in inferior germination. Abnormal seedlings exhibited an absence of lateral roots and true leaf formation (Figure 9). After depletion of nutrients in cotyledons, cotyledons tended to be shriveled. We found that larger-sized seeds performed better (germination, FW, FWPP) than non-sorted seed lots or lower-sized seeds. Other researchers had also linked seed size to crop performance, with factors such as germination and seedling quality [21,22]. Lettuce seeds were reported to have a positive relationship between seed size and germination percentage and seedling vigor [28]. Based on our findings, we also believe that hemp seed lots with more uniform size distribution would have greater yield and quality than non-sorted seed lots. This is because seeds of different size tend to have more variable germination which would then lead to disparities in size and performance, such as larger seeds emerging earlier and shading smaller seeds. For commercial growers, we recommend sieving and discarding the smallest portion of seeds (less than 2.98 mm width) in order to maximize the yield. A limitation of seed lot effect should be considered in the future research. We used the same seed lot with three crop cycles in the seed size experiment, so the effect of seed size on plant performance could also be affected by the seed lot effect. Further studies should be conducted with the hemp seed size and seedlings to understand the effect of seed size on seedling quality (leaf area or stem thickness), nutrient level (phytochemical or mineral nutrients) and interaction with fertilizers (whether large-sized seeds may require less fertilizer to obtain the optimal yield).

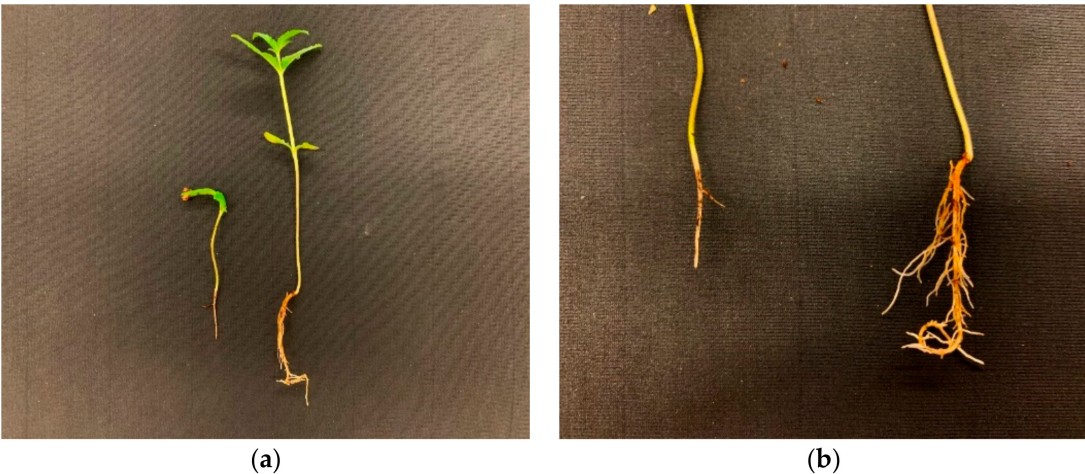

(**a**)                    (**b**)

**Figure 9.** Images of abnormal 'Picolo' seedling (**a**) and as compared to a healthy seedling (**b**).

**5. Conclusions**

The success of baby leaf hemp production requires optimal environmental conditions and cultural management. Based on the results of our studies, and the seed lots we have access to, we recommend growers consider 'Picolo', 'Ferimon', 'X-59', and 'Katani' as productive baby leaf hemp cultivars with high germination percentage that exhibit relatively high yield potential. Similarly, other field hemp cultivars available on the market may be suitable for baby leaf production if they are disease-free and have a high germination rate. Although the sensory analysis and nutrition assessment has not been studied yet, these cultivars could bring the optimal production (fresh weight) for growers to ensure

their profit. For commercial sowing density, we recommend 2.3 to 2.85 seeds·$cm^{-2}$ when harvested at the stage of emergence of the third true leaf. If harvested at an earlier or later development stage, alternative seed densities would need to be studied. Although there was a slight yield benefit with greater than 2.3 seeds·$cm^{-2}$, there was increased incidence of disease observed at the highest density, especially in some growing conditions (high relative humidity or poor airflow). Finally, we found that seed size affected germination and yield in baby leaf hemp production. Around 3% to 25% yield increase could be achieved by sieving and discarding the smallest portion of seeds (<2.98 mm), depending on cultivar (Figure 8). In this research, cultural production methods of baby leaf hemp were developed as a foundational study, but the understanding of baby leaf hemp growing method is still limited. The effect of other management techniques, including light quantity, light quality, $CO_2$ enrichment and fertilizer practices, should be studied for baby leaf hemp in future work.

**Author Contributions:** Investigation, experimental design, data collection and analysis, writing and original draft preparation, R.M.; conceptualization on all aspects of the project, data interpretation and editing the manuscript, N.S.M.; conceptualization on seed size study, data interpretation and editing the manuscript, A.G.T.; conceptualization on variety selection, data interpretation and editing the manuscript, L.B.S. All authors have read and agreed to the published version of the manuscript.

**Funding:** This research received no external funding.

**Acknowledgments:** This material is based upon work that is supported by the United States Hatch Funds under Multi-state Project W-4168 under accession number 1007938 to the second author. This work was supported by New York State Department of Agriculture and Markets through grants AC477 and AC483 from Empire State Development Corporation.

**Conflicts of Interest:** The authors declare no conflict of interest.

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
