# Peer review of "Developing Production Guidelines for Baby Leaf Hemp (Cannabis sativa L.) as an Edible Salad Green: Cultivar, Sowing Density and Seed Size"

_agriculture, doi:10.3390/agriculture10120617_

Round 1

Reviewer 1 Report

Introduction are too long. The sowing density of microgreens is too widely discussed.

Names of varieties should not be repeated in the text of Materials and Methods (page3) as they are given in the Table1.

Do not repeat in each section (2.1; 2.2; 2.3) when the crop was harvested, what was investigated (germination percentage and so on).

In section 2.4 the first three sentences are repeated, this is mentioned in Materials and Methods

It should not be indicated to each graph how the statistics were performed, this is described in section 2.4 of Materials and Methods

De Meijer, 1995; Salentin et al., 2015 (page 2), Mi, 2020 (page 17) are not mentioned in the references.

Not all references are presented according to the requirements: journal titles are not abbreviated and so on. For example, 9 literature sources - the abbreviation of the journal title "Ceylon journal of science" is "Ceylon J. Sci.". For example, 27 literature sources cited incomplete “Progressive Agriculture 1970, 22 (6).”

In the Discussion of the Cultivar Selection Experiment few literature sources were cited, too much general discussion.

Conclusions are too long. Do not repeat what has been studied, but to state what has been established.

Reviewer 2 Report

The authors submitted a paper aiming to find/optimize cultural practices for baby leaf hemp production. The technical hemp (Cannabis sativa L.) attracts recently improved interest due to novel possibilities of fiber processing in industry (recyclable composite materials), but also due to its nutritional and medicinal utilization. One of the recent promising and required nutritional use is a production of edible salad green. The main demand of growers is to obtain guidelines for baby leaf hemp in controlled environment agriculture facilities. The authors concentrated on three components affecting the yield, namely cultivar selection, sowing density and seed-size effect. The experiments were clearly and logically designed, well performed, obtained data were correctly analyzed and interpreted. The obtained data and recommendations fill up the white places in this field and will be very valuable for commercial growers/producers of baby leaf hemp. From the point of view of reviewer, the manuscript may be considered to published with minor corrections and text editing (e.g. citation in line 361 should be replaced by number 28; Fig. 9 should be replaced – if possible – by better quality one etc.).

Author Response

Thanks for your valuable suggestions!

I have already revised the citation you mentioned. However, for the Fig.9, it is a little bit hard for me to find a better picture now. Sorry for the imperfection.